# RARE: Retrieval-Aware Robustness Evaluation for Retrieval-Augmented Generation Systems

## Abstract

Retrieval-Augmented Generation (RAG) enhances recency and factuality in answers. However, existing evaluations rarely test how well these systems cope with real-world noise, conflicting between internal and external retrieved contexts, or fast-changing facts. We introduce **R**etrieval-**A**ware **R**obustness **E**valuation (**RARE**), a unified framework and large-scale benchmark that jointly stress-tests query and document perturbations over dynamic, time-sensitive corpora. One of the central features of RARE is a knowledge-graph-driven synthesis pipeline (RARE-Get) that automatically extracts single and multi-hop relations from the customized corpus and generates multi-level question sets without manual intervention. Leveraging this pipeline, we construct a dataset (RARE-Set) spanning 527 expert-level time-sensitive finance, economics, and policy documents and 48295 questions whose distribution evolves as the underlying sources change. To quantify resilience, we formalize retrieval-conditioned robustness metrics (RARE-Met) that capture a model's ability to remain correct or recover when queries, documents, or real-world retrieval results are systematically altered. Our findings reveal that RAG systems are unexpectedly sensitive to perturbations. Moreover, they consistently demonstrate lower robustness on multi-hop queries compared to single-hop queries across all domains.

## 1 Introduction

Retrieval-Augmented Generation (RAG) significantly enhances Large Language Models (LLM) by integrating external knowledge sources, allowing the generation of accurate and contextually rich responses (Gao et al., 2024). However, the robustness of RAG systems remains inadequately evaluated. In addition, current benchmarks predominantly rely on static, time-invariant datasets with general-knowledge or common-sense queries. Such benchmarks inadvertently favor models that rely on memorization rather than genuine retrieval and synthesis of novel, timely information (Xu et al., 2024). Consequently, existing assessments yield overly optimistic performance measures, overlooking critical real-world scenarios involving dynamic, specialized, and complex information.

An ideal synthesized evaluation dataset generation pipeline for RAG must address several critical dimensions simultaneously, emphasizing **dynamics**, **query complexity**, and **content specialization**. Dynamics is crucial to reflect real-world scenarios where information evolves rapidly (Meem et al., 2024; Jang et al., 2022), particularly in domains such as finance (Shen & Kurshan, 2023). Such time-sensitive data sets prevent contamination of memorized responses and require continuous adaptation by RAG systems. Query complexity, especially multi-hop scenarios that require complex reasoning and integration across multiple retrieved documents (Yang et al., 2018; Geva et al., 2021). Most existing multi-hop datasets require substantial human efforts, which makes it impossible to curate large-scale extensive datasets. Therefore, automation is essential and advanced techniques such as Knowledge Graphs (KGs) (Schneider et al., 2022) can be used. Moreover, with widespread integration into real-world applications, benchmarks must emphasize content

specialization, including professional and domain-specific contexts that challenge models with intricate terminology and nuanced interpretations.

Additionally, most RAG benchmarks has focused on accuracy measurements, with limited attention to how these systems perform when faced with noisy or imperfect inputs. In real-world applications, an RAG system usually should contend with perturbed queries containing typos, irrelevant information, or ambiguous phrasing (Zhang et al., 2025b). Retrieved document may also be noisy, partially relevant, or even contradictory (Chen et al., 2023). A truly robust RAG system should maintain robust performance despite these challenges.

In this paper, we introduce a comprehensive **R**etrieval-**A**ware **R**obustness **E**valuation (**RARE**) framework. It includes: **RARE-Get**: a novel dynamic synthesis pipeline that automatically constructs time-sensitive RAG evaluation data through knowledge graph triplet extraction and traversal techniques, enabling the creation of single-hop and multi-hop tuples (question, answer, ground truth chunks) at various complexity levels without manual curation. **RARE-Set**: a large-scale benchmark comprising 527 specialized documents and 48295 queries across financial, economics, and policy domains - sectors where information accuracy and timeliness are particularly critical yet underrepresented in existing benchmarks. Unlike previous datasets dominated by general knowledge questions, our benchmark exclusively focuses on "rare" datasets: domain-specific, technical queries that require advanced information synthesis. **RARE-Met**: a comprehensive robustness evaluation metric for measuring RAG system performance under perturbations to queries, documents, and simulated real-world retrieval results, providing diagnostic insights into current system limitations. Our dataset features diverse query patterns generated through knowledge graph traversal, including single-hop, multi-hop chained, star-shaped, and inverted-star-shaped, with systematic perturbations at both surface and semantic levels to comprehensively assess robustness under realistic conditions.

Our evaluation reveals that RAG systems are still fragile under some perturbations. Robustness scores do not always scale strictly with model size - some mid-sized generators outperform several larger counterparts. Also, the robustness of RAG systems across different domains is different, and multi-hop queries prove less robust than single-hop queries. All of these indicate the importance of evaluating and improving the robustness of RAG systems.

## 2 RELATED WORK

**Time-Sensitive Benchmark**   Recent temporal-related benchmark initiatives address LLM knowledge outdating through distinct approaches. FreshQA (Vu et al., 2024) tests reasoning over up-to-date knowledge with a fixed questions, dynamic answers-updated QA benchmark and evaluation methodology for correctness and hallucination detection. PAT-Questions (Meem et al., 2024) introduces a self-updating benchmark for present-anchored temporal questions using SPARQL queries over Wikidata to automatically refresh answers. RealtimeQA (Kasai et al., 2024) employs a weekly dynamic platform that extracts questions form news quizzes, challenging systems to answer questions about current events. Existing benchmarks often exhibit limitations such as narrow raw data domains (primarily Wikipedia or news articles), a restricted number of test cases due to the reliance on fixed human-generated questions, and a prevalence of queries that can be accurately answered by the language model alone—without the need for retrieval—such as general-domain fact questions.

**Multi-Hop QA and RAG Benchmark**   Early knowledge-intensive benchmarks like Natural Questions (Kwiatkowski et al., 2019) and HotpotQA (Yang et al., 2018) established foundations for factual question answering but lacked cross-document reasoning and overlapping with popular training dataset. Later development such as MuSiQue (Trivedi et al., 2022) and StrategyQA (Geva et al., 2021) advanced multi-hop QA capabilities but remained confined to Wikipedia sources. MultiHop-RAG (Tang & Yang, 2024) expanded to news domain with 2-4 ho queries but lacks dynamic real-time updates. RAGBench (Friel et al., 2025)

Table 1: Comparison of our proposed dataset with prior benchmarks. Symbols: ✓ = yes/present; ✗ = not available; "partial" = feature applies to only a subset; "-" = not applicable; MH = Multi Hop question.

| Dataset | Year | # QA | Data Sources | Unique | Time-Sens. | MH | Dynamic | Automatic |
|---|---|---|---|---|---|---|---|---|
| **Time-Sens. Benchmarks** | | | | | | | | |
| RealtimeQA | 2023 | 2340 | News | ✓ | ✓ | ✓ | ✓ | partial |
| FreshQA | 2024 | 600 | Search engine | ✓ | ✓ | ✓ | ✓ | partial |
| PAT-Questions | 2024 | 6172 | Wikipedia | partial | ✓ | ✓ | ✓ | ✓ |
| **MH & RAG Benchmarks** | | | | | | | | |
| Natural Questions | 2019 | 100 k | Wikipedia | ✗ | ✗ | ✗ | ✗ | ✗ |
| HotpotQA | 2018 | 97.9 k | Wikipedia | ✓ | ✗ | ✓ | ✗ | ✗ |
| MuSiQue-Ans | 2022 | 50 k | Wikipedia | ✗ | ✗ | ✓ | ✗ | partial |
| StrategyQA | 2021 | 2780 | Wikipedia | ✓ | ✗ | ✓ | ✗ | ✗ |
| MultiHop-RAG | 2024 | 2506 | News | ✓ | ✓ | ✓ | ✗ | ✓ |
| RAGBench | 2024 | 100 k | Domain-specific | ✗ | ✗ | ✓ | ✗ | ✓ |
| CRAG | 2024 | 4409 | Search engine | ✗ | ✓ | ✓ | ✗ | partial |
| **LLM Robust Benchmarks** | | | | | | | | |
| KaRR | 2023 | - | T-REx (Wikipedia) | partial | ✗ | ✗ | ✗ | partial |
| QE-RAG | 2025 | 51 k | Wiki + Domain-specific | partial | ✗ | ✓ | ✗ | ✓ |
| SURE | 2025 | - | NQ-open (Wikipedia) | ✗ | ✗ | ✗ | ✗ | ✓ |
| **RARE (Ours)** | 2025 | 48.3 k | Domain-specific reports | ✓ | ✓ | ✓ | ✓ | ✓ |

introduced evaluation across industry corpora with new faithfulness metrics, with CRAG (Yang et al., 2024) targets dynamic performance across multiple domains with simulated web and knowledge graph APIs, though still limited in scale and dynamic renew ability.

**LLM & RAG Robustness**  Recent frameworks attempt to quantify RAG robustness, usually with various perturbations. RAGAS (Es et al., 2025) measures factual consistency through automated evaluation without ground-truth annotations but lacks assessment of query/document perturbations and limited number of assessment. Cao et al. (2025) analyzed the robustness of the RAG system on linguistic variations and found that RAG systems are even more sensitive to these variations compared with LLM-only generation. SURE Yang et al. (2025b) introduced a framework to quantify the sensitivity to semantic-agnostic spurious features (e.g. format of document) in grounding data, providing a taxonomy of formatting variations that reveal widespread vulnerabilities. QE-RAG (Zhang et al., 2025b) tests robustness by injecting realistic query entry errors into QA datasets to evaluate tolerance to input noise, though primarily focused on static, general-domain tasks without evaluating document-level corruptions. KaRR (Dong et al., 2023) provides a statistical approach to assess whether an LLM contains reliable factual knowledge by estimating the ration of generating correct surface text given varying prompts, although its assessment is limited to parametric knowledge rather than retrieval capabilities. While these approaches advance discrete facts of RAG robustness, none offer a unified, dynamic evaluation pipeline capable of automatically generating large-scale, temporal test cases and measuring performance under systematic perturbations to queries, documents, and retrieval results.

## 3 RARE-GET: Dynamic RAG Benchmark Dataset Generation Pipeline

RAG benchmarks should ideally comprise diverse, realistic queries with corresponding golden passages containing the information needed to answer them correctly. Creating such benchmarks manually demands extensive human effort and domain expertise, particularly for specialized, multi-hop reasoning scenarios. In addition, manual-based benchmark cannot consistently create the dynamic and up-to-date datasets. To address these challenges, we introduce RARE-Get, a fully automated pipeline for constructing complex RAG benchmarks from unstructured data.

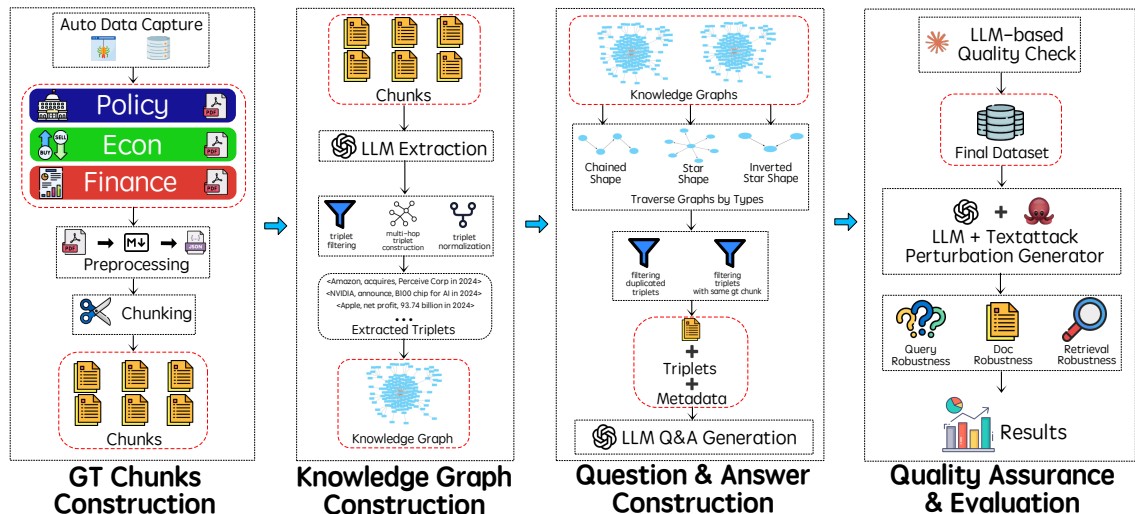

Figure 1: Illustration for the RARE framework. Red frame: data that pipeline will generate; **Black frame**: process/movement.

RARE-Get transforms domain-specific documents into comprehensive benchmark datasets through four key stages: (1) Ground Truth Chunks Construction; (2) Knowledge Graph Construction; (3) Question & Answer Construction and (4) Quality Assurance, as illustrated in Figure 1. This approach enables the creation of technical, challenging RAG evaluation datasets that evolve dynamically alongside their source documents, ensuring continued relevance in rapidly changing domains. For time-sensitive, such automatic pipeline also ensures that newest answers with questions will always be updated following by the knowledge graph re-construction or updating process.

## 3.1 CORPUS PREPARATION AND CHUNKING

The pipeline begins by processing domain-specific documents, converting them into manageable chunks suitable for retrieval systems. We carefully segment each document into passages of approximately 600 tokens, striking a balance between informativeness and retrieval efficiency, as well as a real-world retrieval simulation. For tables, we prevent splitting a single table across different chunks. Related information (e.g. table titles, data explanation) will remain in the same chunk. Similarly, for text-only contents, we ensure that no paragraph is divided between chunks. Also, we develop specialized chunking techniques across three distinct domains. Each domain receives tailored processing to enhance information extraction and context retention. Appendix A illustrates the full details for chunking on different domains.

## 3.2 KNOWLEDGE GRAPH EXTRACTION

The cornerstone of the benchmark creation process is systematically transforming chunked documents into structured knowledge representations. For each set of $n$ consecutive chunks, we employ LLM (GPT-4.1 (OpenAI, 2025)) with carefully designed prompts adapted for different domains. The prompts specify multiple types of multi-hop question patterns with detailed examples, instructing the LLM to extract connected triplets where entities overlap between chunks. In addition, we require the LLM to output the exact source sentence used to extract each triplet; this sentence is later used for validation through a normalized exact-match check to ensure that all extracted relations are fully grounded in the originating chunk, discarding any triplets whose sources are unverifiable.

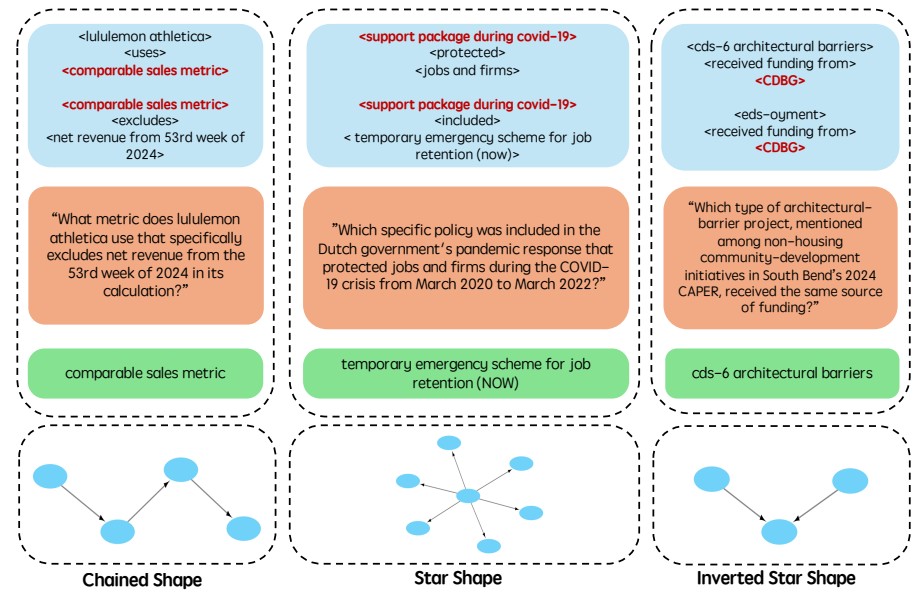

Figure 2: Examples of the multi-hop questions. Blue: triplets traversed from KG; Peach: generated question; Green: generated answer; Red: "bridge" entity which connect different triplets together;

To ensure the consistency of the knowledge graph label, we further normalize semantically similar relations (e.g., "manufactures" vs. "produces") using E5-Mistral-7B-Instruct (Wang et al., 2023), one of the leading embedding models according to the MTEB leaderboard (Muennighoff et al., 2023). New relation labels are mapped to existing relation when their cosine similarity exceeds a threshold, which we selected as 0.9 based on human majority voting over sampled relation pairs.

We also perform entity alignment using a text-normalization procedure (lower-casing, punctuation removal, and domain-specific stripping of corporate suffixes such as "Inc." or "Corp." in finance). Finally, after constructing the knowledge graph for each individual document, we merge the per-document graphs into a larger cross-document knowledge graph using NetworkX (Hagberg et al., 2008) to support multi-hop and cross-file question generation. Example prompts used for the extraction of triplets are provided in Appendix F.

## 3.3 QUERY PATTERNS

By traversing the constructed knowledge graph in different strategies, we identify four structural templates, one single-hop and three multi-hop, that produce queries of increasing complexity (multi-hop examples and QA pairs appear in Figure 2).

When traversing the entire graph according to these patterns and identifying the corresponding triplet(s), we ensure that the extracted triplets can only be used to generate corresponding questions. For instance, while traversing all single-hop triplets $(e_1, r_1, e_2)$, we ensure that $e_1$ has an out-degree of 1 and an in-degree of 0, while $e_2$ has an in-degree of 1 and an out-degree of 0. This approach prevents duplication of content between single-hop and multi-hop questions. Additionally, for multi-hop questions, we remove all triplet sets that can be entirely answered from the same chunk. This ensures that multi-hop questions must be answered by traversing multiple files. Finally, We restrict to these patterns because they cover the three fundamental reasoning moves in real retrieval: follow a path (chain), aggregate around a hub (star), and converge multiple

| Pattern | Graph structure (template) | What it tests |
|---------|----------------------------|---------------|
| Single-hop | $(e_1, r_1, e_2)$ | Direct fact lookup; baseline single-chunk retrieval. |
| Chained-Shape | $(e_1, r_1, \boldsymbol{e_2}) \rightarrow (\boldsymbol{e_2}, r_2, e_3) \rightarrow \ldots$ | Follow 2–3 linked triplets; step-wise reasoning across chunks. |
| Star-Shape | $(\boldsymbol{e_1}, r_1, e_2) \parallel (\boldsymbol{e_1}, r_2, e_3) \parallel \ldots$ | Aggregate diverse facts around a focal entity; synthesize across relations. |
| Inverted-Star | $(e_1, r_1, \boldsymbol{e_2}) \parallel (e_3, r_2, \boldsymbol{e_2}) \parallel \ldots$ | Recognize convergent paths; combine evidence toward a common target. |

Table 2: Single hop and multi-hop query pattern templates.

clues to a target (inverted-star). These patterns are expressive enough to span most cross-chunk tasks while keeping graph traversal depth and branching controllable for automatic generation, verification, and difficulty tuning.

### 3.4 Query Generation and Quality Assurance

For each identified pattern, we use pattern-specific prompts to generate QA pairs that use information from its triplets, corresponding ground truth chunks, and metadata storing information such as timestamp or the country name. For multi-hop questions specifically, we implement a specialized algorithm that: (1) Identifies a "pivot entity" that connects different triplets; (2) References this pivot indirectly in the question; (3) Ensures the question cannot be answered from a single chunk; (4) Performs "pivot-rarity" and "negative-distractor safety" checks to guarantee question quality. Appendix F shows the complete algorithm for generating pairs.

Finally, all generated query-answer pairs undergo rigorous quality assessment using separate LLM-based evaluation based on Claude 3.5 Haiku (Claude, 2024) that scores each query-answer pair on three dimensions from the scale of 1 to 5: (1) Readability; (2) Clarity; (3) Correctness. Only queries with scores above 3 across all dimensions are included in the final benchmark. This quality-controlled generation process creates benchmarks that effectively evaluate both retrieval accuracy and reasoning capabilities within domain-specific contexts. As source documents evolve or new ones are added, the pipeline can dynamically extend the benchmark, ensuring continued relevance for evaluating RAG systems against the latest information. Appendix F includes step-by-step measuring standards.

## 4 RARE-Set: Large-Scale Domain-Specific RAG Dataset

RARE-Set contains three different domains of datasets: finance, economics, and policy. We collect a heterogeneous corpus with 199 recent S&P 500 Companies' SEC 10-k filings, 114 OECD economic surveys, and 214 Consolidated Annual Performance and Evaluation Report (CAPER) from grantees for U.S. Department of Housing and Urban Development (HUD) funded programs. Appendix E shows the full dataset statistics.

We enhance datasets quality through a variety of processing techniques. For instance, for financial reports, our preprocessing pipeline builds on Edgar-Crawler (Loukas et al., 2021), with custom modifications. Rather than preserving tables in HTML format, we convert them to a markdown structure optimized for LLM inputs. In knowledge graph extraction from financial documents, we prioritize relations involving performance metrics, operational activities, and financial events. We explicitly target generalized and reusable relations that can be applied across companies within the same industry. This approach supports the generation of multi-hop questions that span multiple companies. For economic surveys, we design prompts to emphasize policy measures, key economic indicators, and patterns of national development. In the context of policy reports, our focus is on fund allocation, program implementation, and beneficiary data.

Table 3: Robustness definitions under query/document settings. ✓ = counted robust only if the final answer is correct; ∅ = counted robust only if the model safely refuses; ✓ ∨ ∅ = robust if either correct or safely refuses. $g(q, d)$ represents generator (model) given query and document. $g(q, \emptyset)$ is the per-record no-context probe indicating the generator can answer without retrieval. 1 denotes that the generator can answer without retrieval, while 0 indicates it cannot.

| | $g(q, \emptyset) = 1$ | | $g(q, \emptyset) = 0$ | |
| --- | --- | --- | --- | --- |
| **Document setting** | $q$ (orig.) | $q'$ (perturbed) | $q$ (orig.) | $q'$ (perturbed) |
| Ground-Truth Docs | ✓ | ✓ | ✓ | ✓ |
| Lexical-Diff (Has Answer) | ✓ | ✓ | ✓ | ✓ |
| Lexical-Similar (No Answer) | ✓ ∨ ∅ | ✓ ∨ ∅ | ∅ | ∅ |
| Real-World Retrieval | ✓ ∨ ∅ | ✓ ∨ ∅ | ✓ ∨ ∅ | ✓ ∨ ∅ |

The benchmark contains single-hop queries and three types of multi-hop queries based on different knowledge patterns in the knowledge graph. One thing to mention is that all of these domains are time-sensitive and can update dynamically as time progresses.

Finally, to evaluate the factual accuracy of the extracted knowledge-graph triplets relative to their corresponding source chunk, we randomly sampled 1000 triplets and assessed them using Claude Sonnet 4.5 (since GPT-4.1 generated the triplets). The evaluation indicates that 87.1% of the sampled triplets are factually correct with respect to their originating chunks, suggesting that the triplets effectively preserve information from the source chunks. The verification prompt is in Appendix F. Moreover, in the LLM-based quality-assurance stage, only 19.69% of the generated QA pairs were filtered out.

## 5 RARE-MET: RETRIEVAL-AWARE ROBUSTNESS METRIC

A robust RAG system should maintain correctness under two conditions: if the generator can already answer the query without retrieval ($g(q, \emptyset) = 1$), it must consistently give the correct answer regardless of retrieval content; if it cannot answer without retrieval ($g(q, \emptyset) = 0$), it should provide the correct answer given correct retrieval, and otherwise safely refuse rather than hallucinate when retrieval is incorrect or irrelevant.

Table 3 shows the full definition of RAG robustness under different circumstances.

### 5.1 QUERY PERTURBATIONS

We define four types of query perturbations $Q' = q'_1, q'_2, \ldots, q'_n$ derived from the original query $q$, grouped into two categories: **Surface-level perturbations**: (1) character-level changes; (2) word-level changes (typos, synonyms) based on TextAttack (Morris et al., 2020); and **Advanced-level perturbations**: (1) LLM-based grammar rewrites that preserve the query's intrinsic meaning; (2) LLM-based additions of irrelevant information. Appendix C.1 includes more details on constructing perturbations for each query.

### 5.2 DOCUMENT PERTURBATION

For document perturbation $D' = d'_1, d'_2, \ldots, d'_n$, we primarily consider two directions: lexical relevance and answer relevance. Similarly to definitions under query perturbation, the lexical relevance measure changes of document styles. Answer relevance, on the other hand, determines whether the retrieved document truly contains the answer required by the question. As we consider lexical perturbation and answer perturbation as two dimensions, we define three document perturbations which encompassed all possible distributions of retrieval documents. (1) Documents with the similar lexical style but answers are different: directly remove the answer sentence/words from the ground truth chunk. (2) Documents with different lexical style but

answer is similar/identical: LLM-based back-translation. (3) Real-world retrieval results ($D_{ret}$): constructing a real-world simulated retrieval process based on LangChain (Chase & contributors, 2022) (including a re-ranking model). The first two document perturbations are introduced to more clearly examine how different relevance types—lexical or answer-based—affect the overall robustness of the RAG system.

Appendix B shows all types of document perturbations under such relevance and reason of evaluating from these perspectives. Appendix C.2 reveals construction process in details. The first two document perturbations are introduced to more clearly examine how different relevance types—lexical or answer-based—affect the overall robustness of the RAG system.

## 5.3 ROBUSTNESS METRICS

| Metric | Fixed / Varied | Expression |
|---|---|---|
| Overall Robustness | Fixed: $\emptyset$; Varied: $q \in Q,\ d \in D$ | $\dfrac{1}{\|Q\|\,\|D\|}\sum_{q \in Q}\sum_{d \in D} f\big(g(q,d),a\big)$ |
| Query Robustness | Fixed: $d_{\text{gt}}$; Varied: $q' \in Q'$ | $\dfrac{1}{\|Q'\|}\sum_{q' \in Q'} f\big(g(q',d_{\text{gt}}),a\big)$ |
| Document Robustness | Fixed: $q$; Varied: $d' \in D'$ | $\dfrac{1}{\|D'\|}\sum_{d' \in D'} f\big(g(q,d'),a\big)$ |
| Real-World Retrieval Robustness | Fixed: $q$; Varied: $d_i' \in D_{\text{ret}}$ | $\dfrac{1}{\|D_{\text{ret}}\|}\sum_{d_i' \in D_{\text{ret}}} f\big(g(q,d_i'),a\big)$ |

Table 4: Definition of different robustness score. $f(pred,\ ans)$ indicates the open-ended prediction and ground truth LLM-based comparison function. All other notations are identical to the previous section. Appendix D also provides an additional table to understand these notation better.

# 6 ROBUSTNESS EXPERIMENTS AND ANALYSIS

## 6.1 EXPERIMENTAL SETTING

We perform our experiments on a total of 6000 QA pairs for three domains, each of which has 1000 single-hop questions and 1000 multi-hop questions. Retrieval is evaluated with three top-ranking embedding models from the MTEB leaderboard: E5-Large-Instruct Es et al. (2025), Jina-Embedding-v3 Sturua et al. (2024), and Stella-En-1.5B-v5 Zhang et al. (2025a). For the RAG system's generators, we evaluate both leading open-source LLMs, including Qwen 3 Yang et al. (2025a) and the Llama 3.2 family Grattafiori et al. (2024), as well as proprietary models accessed through commercial APIs. The Llama 3.2 series is served via the Amazon Bedrock API, while closed-source GPT models are accessed directly through the OpenAI API. Our total expenditure on the GPT-4.1 series models was approximately $3400, which includes costs for KG triple extraction (GPT-4.1 only), QA generation (GPT-4.1 only), and evaluations involving three different GPT models. All generators are configured to operate deterministically (temperature = 0) with a maximum output length of 1024 tokens. Although models are instructed to provide concise final answers, chain-of-thought reasoning is explicitly encouraged in their outputs to facilitate their abilities. We close Qwen 3's internal thinking mode for fair comparison. Appendix G.3 proves our results are statistical significance.

For the Qwen 3 series, we deploy both vLLM Kwon et al. (2023) servers (for larger models) and SGLang Zheng et al. (2024) servers (for smaller models), running in parallel with their official recommended settings to optimize inference throughput and performance. These open-source models are executed on a cluster of 16 NVIDIA L40S GPUs. To accelerate large-scale experimentation, multiple server instances are launched

concurrently, and inference requests are distributed across them. Completion of the full experimental suite requires approximately five days.

To quantify the discrepancy between predictions and ground-truth answers, we design a two-stage evaluation pipeline. In the first stage, both prediction and reference strings are normalized, after which exact and inclusive string matches are implemented. If no lexical match is detected, the second stage employs Claude-3-Haiku Anthropic (2024) judging with a carefully engineered evaluation prompt (Appendix F) to determine whether prediction matches the ground truth. Using Claude model can minimize bias and ensure neutrality in the evaluation.

Table 5: Robustness results across different models and metrics

| Model | Overall | Query | Document | Retrieval |
|---|---|---|---|---|
| Llama-3.2-1B-Instruct | 0.318 | 0.280 | 0.254 | 0.389 |
| Llama-3.2-3B-Instruct | 0.607 | 0.459 | 0.587 | 0.649 |
| Llama-3.2-11B-Vision-Instruct | 0.627 | 0.658 | 0.630 | 0.610 |
| Llama-3.2-90B-Vision-Instruct | **0.782** | 0.691 | **0.771** | **0.820** |
| Qwen3-4B | 0.665 | 0.734 | 0.700 | 0.611 |
| Qwen3-8B | 0.698 | 0.714 | 0.721 | 0.667 |
| Qwen3-32B | 0.664 | 0.732 | 0.701 | 0.598 |
| GPT-4.1-nano | 0.589 | 0.613 | 0.651 | 0.531 |
| GPT-4.1-mini | 0.646 | 0.730 | 0.651 | 0.613 |
| GPT-4.1 | 0.675 | **0.761** | 0.668 | 0.654 |

## 6.2 OVERALL ROBUSTNESS TRENDS ACROSS MODEL SCALES

Examining the overall robustness scores in the Table 5 shows that larger models generally demonstrate superior robustness. GPT-4.1 achieves a robustness score that surpasses those of its smaller models, GPT-4.1 mini and GPT-4.1 nano. A similar scaling-law is observed within the Llama 3.2 series: Llama-3.2-90B-Vision-Instruct exhibits a markedly higher robustness score than any other model. Surprisingly, it even surpasses closed models such as GPT series. However, size alone does not always reflects the robustness. For example, Qwen3-32B attains an overall robustness score lower than that of the smaller, but architecturally similar Qwen3-8B and even Qwen3-4B. This phenomenon is widely observed across the Qwen3 family of models. The Qwen3 models consistently maintain a relatively high robustness score, even for smaller-scale variants such as the 4B model. In addition, compared with other robustness scores, the document score does not exhibit a significant improvement as model size increases; in fact, some models even show regression.

This is primary because some larger models are more likely to answer directly with hallucinations, even when they lack the ability to answer the question or when the given document does not contain the answer. However, certain smaller models are more likely to decline questions that exceed their capabilities. As shown in Figure 9, while the ground-truth and lexical-different-with-answer robustness scores generally follow the scaling law, the other two types of document robustness do not, especially the lexical-similar-without-answer robustness. Smaller models typically achieve higher scores in this sub-score due to their higher probability of issuing refusals. Larger models tend to respond to the question more frequently than smaller models, which leads to lower lexical-similarity robustness scores. This behavior ultimately affects the document robustness as well as the overall robustness score.

In contrast, Figure 8 shows that there are no significant differences across different query perturbations within each model, indicating that current models exhibit similar consistency when facing various query perturbations. Across models, the query robustness scores generally follow the scaling law, with Qwen3 models consistently achieving high scores.

### 6.3 DOMAIN-SPECIFIC AND MULTI-HOP QUESTIONS ROBUSTNESS

Figure 7 indicates that the robustness of RAG systems is heavily influenced by domain-specific factors. RAG system perform best in finance reports, which typically feature standardized terminology and numerical data. However, they are struggling most with the economics survey, which often involves complex causal relationships and varied terminology. In addition, single-hop queries consistently yield higher robustness scores than multi-hop queries across all domains and perturbations (Figure 6). This trend is amplified in smaller models, suggesting that multi-hop reasoning capabilities require substantial model capacity to maintain robustness under perturbations.

Appendix G includes all evaluation results.

## 7 CONCLUSION

In conclusion, we introduce RARE, a comprehensive framework for data generation and evaluating RAG robustness that addresses critical gaps in existing benchmarks. Our knowledge-graph-based pipeline (RARE-Get) automatically extracts relations from specialized corpora and generates multilevel questions through pattern-based traversal, enabling dynamic dataset evolution without manual curation. The resulting benchmark (RARE-Set) comprises 48295 questions across finance, economics, and policy domains, featuring single-hop and complex multi-hop questions. Our robust evaluation metrics (RARE-Met) systematically measure resilience against query, document, and retrieval perturbations. Experiments reveal that RAG systems consistently demonstrate higher robustness in finance than economics, and single-hop queries outperform multi-hop ones across all domains, providing crucial insights for developing more reliable RAG systems for real-world applications.

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

## A  CHUNKING TECHNIQUES

For each domain, here are the step-by-step explanation for chunking.

### A.1  FINANCE

1. Load filing JSON and prepare metadata (CIK, company, filing type/date, period; optional GICS sector/subindustry).

2. Preprocess `item_7`: split by lines, detect section titles (regex on uppercase "Item" patterns), detect table-like blocks (pipe-delimited), group tables with nearby narrative, and merge short title-only segments into adjacent content.

3. For each segment:
   - If it contains a table, emit a single chunk with `contains_table=true`.
   - Otherwise, split text with a token-aware recursive splitter (`chunk_size=800`, `overlap=100`, tiktoken-based length), merge very short fragments ($< 30$ words), and carry the section title into the first chunk; mark `contains_table=false`.

4. Assign chunk IDs and attach metadata.

### A.2  ECONOMICS

1. Load structured content; extract `file_country` and `file_year` from the first "OECD Economic Surveys:" line; initialize per-chunk metadata.

2. Start near the first table ($idx = \max(0, \text{first-table-index} - 1)$) and iterate rows.

3. For text rows, accumulate lines until around 600 words, then flush a `text` chunk with `chunk_page_idx`.

4. For table rows, convert HTML to Markdown, prepend detected caption (from row or preceding short "Table" lines) and append footnotes; emit a `table` chunk with `chunk_page_idx`.

5. Flush any remaining text; assign chunk IDs and attach metadata.

### A.3  POLICY

1. Load structured content and join with external metadata row by `id`; prepare per-chunk metadata (plan type, `file_grantee`, `file_state`, `file_year`).

2. Trim trailing content starting at the first "Attachment" header.

3. For text rows, accumulate lines until around 600 words, then flush a `text` chunk with `chunk_page_idx`.

4. For table rows, convert HTML to Markdown; if captions/footnotes exist, prepend/append them; emit a `table` chunk with `chunk_page_idx`.

5. Flush any remaining text; assign chunk IDs and attach metadata.

## B  THREE TYPES OF DOCUMENT PERTURBATIONS

Figure 3 illustrates that real-world retrieval results (violet dots) are scattered throughout the entire space of lexical relevance and answer relevance, indicating that outcomes can occur in any region depending on the retrieval performance. To study robustness, we introduce document perturbations in two targeted regions:

answer-similar but lexically different (orange) and answer-different but lexically similar (blue), which allow us to isolate and examine the impact of lexical versus answer relevance on RAG system performance.

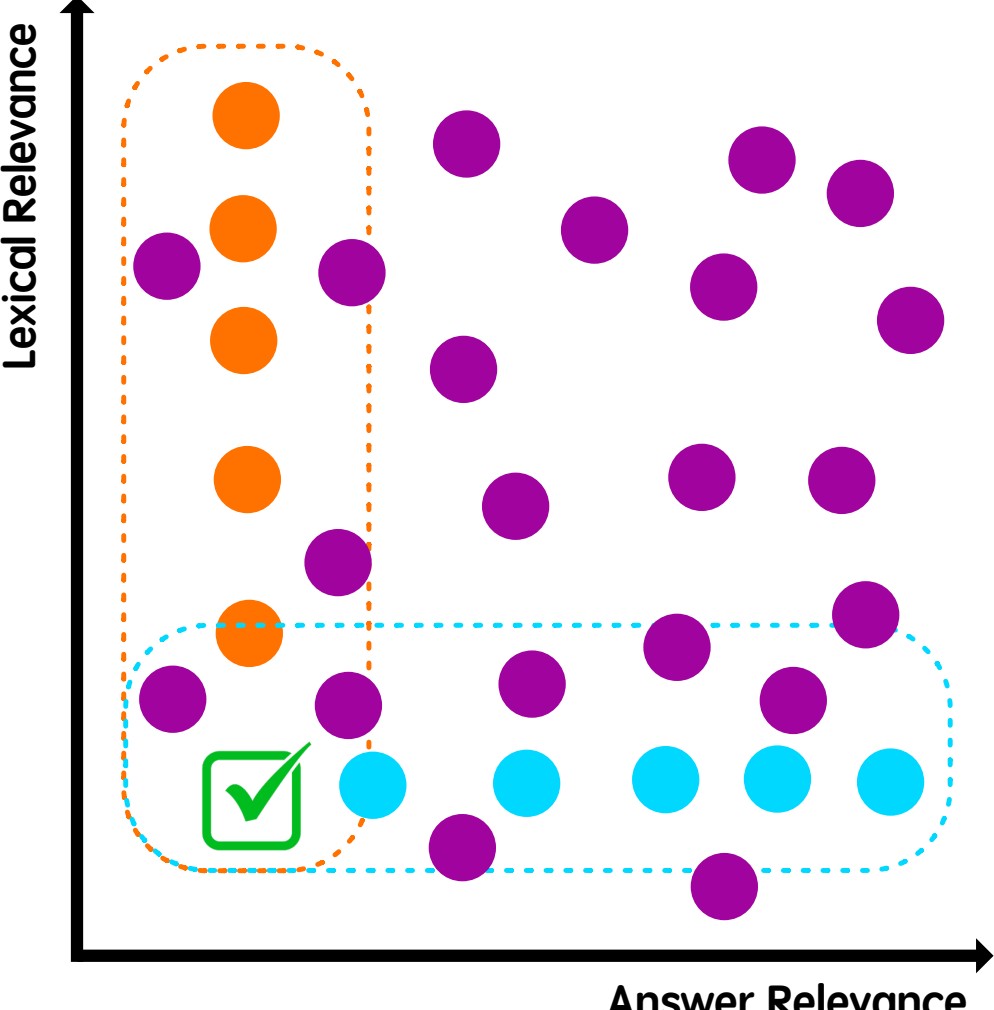

Figure 3: Three types of document perturbations measured by two relevances.

## C  PERTURBATION CONSTRUCTIONS

### C.1  QUERY PERTURBATIONS

1. Character-level noise: Use TextAttack Augmenters such as CompositeTransformation, WordSwapQWERTY and WordSwapRandomCharacterDeletion (swap only 10% of the charac-

ters); sample up to 5 variants and select via embedding model (first passing, otherwise maximum similarity score).

2. Word-level substitutions: Use TextAttack Augmenter with WordSwapEmbedding (max_candidates = 50) and the same constraints; sample up to 5 variants (swap only 10% of the vocabulary) and select with the same embedding model similarity filter.

3. Insert irrelevant info (LLM): Use GPT-4.1 to rewrite the query by inserting one domain-relevant but answer-irrelevant detail (3 candidates); keep the highest-similarity candidate .

4. Grammar perturbation (LLM): Use GPT-4.1 to rephrase only grammar/punctuation/word order (3 candidates); keep the highest-similarity candidate.

## C.2 DOCUMENT PERTURBATIONS

1. Regex deletion: Use Python `re.sub`, `re.escape` and `re.IGNORECASE` to remove exact supporting sentences from answer-bearing chunks; compute semantic similarity using embedding model to the original chunk, ensuring that their core contents are not changed.

2. Back-translation (LLM): Use GPT-4.1 to translate chunks EN→FR then FR→EN in batch; compute similarity to the original with embedding model and attach the perturbed text with its score.

## D  RARE-MET NOTATION REFERENCE

Table 6: Notations and Definitions

| Notation | Definition |
|---|---|
| $q$ | Original query. |
| $q' \in Q'$ | Perturbed query; $Q'$ is the set of query perturbations. |
| $Q = \{q\} \cup Q'$ | Full query set (original + perturbations). |
| $d_{\text{gt}}$ | Ground-truth document. |
| $d' \in D'$ | Perturbed document; $D'$ is the set of document perturbations. |
| $D = \{d_{\text{gt}}\} \cup D'$ | Full document set (ground truth + perturbations). |
| $\emptyset$ | Empty context (no retrieval). |
| $g(q, d)$ | Generator producing the results given question $q$ with context $d$. |
| $a$ | Ground-truth answer. |
| $f(g(q, d), a) \in \{0, 1\}$ | Robustness judge (1 = robust, 0 = not), following Table 3. |
| $g(q, \emptyset) \in \{0, 1\}$ | Parametric-knowledge probe: 1 = can answer without retrieval; 0 = cannot. |
| $D_{\text{ret}}$ | Set of documents returned by the evaluated retrievers. |
| $d'_i \in D_{\text{ret}}$ | A retrieved document used in real-world evaluation (e.g., top-$k$ per retriever). |

## E    RARE-SET STATISTICS

Table 7: Dataset Statistics by Domain

| Domain | Financial | Economics | Policy |
|---|---|---|---|
| Document | 199 | 114 | 214 |
| Chunk | 19825 | 12915 | 7014 |
| Time Scope | 2024-2025 | 2020-2025 | 2024-2025 |
| **Total # of Eligible Triplet/Triplets** | | | |
| Single-hop | 17585 | 6719 | 6176 |
| Chained (multi-hop) | 11193 | 22256 | 82885 |
| Star-shaped (multi-hop) | 2707 | 1780 | 4868 |
| Inverted-star-shaped (multi-hop) | 558 | 2636 | 7377 |
| **Query (Train)** | | | |
| Single-hop | 7362 | 6715 | 6125 |
| Chained (multi-hop) | 7930 | 3863 | 7563 |
| Star-shaped (multi-hop) | 833 | 511 | 661 |
| Inverted-star-shaped (multi-hop) | 64 | 415 | 253 |
| **Query (Test)** | | | |
| Single-hop | 1000 | 1000 | 1000 |
| Chained (multi-hop) | 687 | 774 | 805 |
| Star-shaped (multi-hop) | 289 | 193 | 135 |
| Inverted-star-shaped (multi-hop) | 24 | 33 | 60 |

## F    PROMPTS

We will use the economic dataset prompts as the example.

---

**Dataset Generation Prompt: Triplets Extraction**

You are an economic analyst skilled at interpreting OECD Economic Surveys.
Your task is to extract structured triplets consisting of {"entity_1", "relation", "entity_2"} from provided consecutive text chunks from a single OECD Economic Survey.
Each triplet must be supported explicitly by one specific chunk, but other chunks can be referenced to form insightful, multi-hop triplets.
You should include the source chunk ID and source sentence as the metadata of the triplets.

**TASK: EXTRACT STRUCTURED MULTI-HOP TRIPLETS**

Extract triplets fitting these multi-hop categories:
- Connected Chain
- Star
- Inverted Star

**1. Connected Chain Triplets:**

- Extract an initial triplet: <entity_1, relation, entity_2>.
- Then identify subsequent triplets where entity_2 of the previous triplet becomes entity_1 of the next.

---

- Ideally, different subsequent triplets should be sourced from different chunks.
- Extract as many meaningful chains as possible.
- Skip if no valid connected chain is available.

*Example:*
- {"entity_1": "Luxembourg", "relation": "implemented", "entity_2": "free public transport"}
- {"entity_1": "free public transport", "relation": "aims to reduce", "entity_2": "carbon emissions"}

**2. Star Triplets:**
- One root entity branching into multiple distinct relationships.
- Each branch must independently derive from a unique chunk.
- Skip if no meaningful star relationship is possible.

*Example:*
- {"entity_1": "Luxembourg", "relation": "invests in", "entity_2": "renewable energy"}
- {"entity_1": "Luxembourg", "relation": "develops", "entity_2": "sustainable transport infrastructure"}

**3. Inverted Star Triplets:**
- Two distinct entities connected through a shared attribute (entity_2).
- Relations may differ and offer varied perspectives on the attribute.
- Skip if no valid inverted star relationship is possible.

*Example:*
- {"entity_1": "Luxembourg", "relation": "faces challenges in", "entity_2": "housing affordability"}
- {"entity_1": "OECD recommendations", "relation": "address", "entity_2": "housing affordability"}

**REQUIRED STRUCTURE:**
Each extracted triplet must include:
- entity_1 (str)
- relation (str)
- entity_2 (str)
- answer_chunk_id (str)
- The chunk ID is at the very beginning of each text chunk, such as "Chunk ID: economics_0e32d909-en_chunk_9".
- You should copy the chunk ID where the triplet is extracted from as the "answer_chunk_id".
- source_sentence (str)
- Extracted exactly from the supporting chunk, COPY WORD BY WORD.
- If sourced from a table, strictly include relevant row, column, and specific data only.

**CRITICAL INSTRUCTIONS:**
*Relations:*
- Generalized and reusable across similar economic and policy contexts.
- Concise and specific (2-4 words preferred).
- Use standard economic and policy terminology.

- Avoid specific dates or overly detailed references in the relations.
*Good Examples:*
- "implemented", "faces challenges in", "invests in", "promotes"
*Bad Examples:*
- "introduced free transport in 2020", "planned reforms announced in 2023"

*Entities:*
- Clearly specify entities (avoid general terms like "the country" or "the government").
- Maintain consistent terminology when referring to similar concepts, such as using "Luxembourg" all the time instead of using "Luxembourg government" sometimes.
- Include specific, detailed information relevant to economic policies, recommendations, or outcomes.
- For table-derived entities, clearly indicate row, column, and description.

**Goal:**
Try to extract 15 to 20 triplets. If no valid connected triplets can be extracted, return an empty array:
[]

---

### Dataset Generation Prompt: Single-Hop QA Pairs Generation

Create an economics-related natural question-answer pair using a relation triplet (entity_1, relation, entity_2) based on the text context and the file metadata where the triplet was extracted.

**Requirements**
- The question and answer should be entirely based on the given text context; that is, one can only generate the correct answer from the information available in the context.
- Always use "{file_country}" instead of "{file_country} government," "government," or "country" to make the query more specific.
- You should use entity_1 or entity_2 as the answer to the question and construct the question using the other entity and relation with appropriate context information.
- Aim to formulate questions that appear natural and are likely to be asked by a human.
- Avoid generating questions that are overly general or vague, where multiple ground truth chunks could answer the question or it would be hard to retrieve the ground truth chunk given the question. You MUST return an EMPTY string for question and answer in this case.

**Examples**
*Example 1:*
Triplet:
{"entity_1": "inflation", "relation": "is", "entity_2": "2.9% in 2023"}
Text Context:
`[Full example context is omitted...]`
Metadata:
- File Type: OECD Economic Surveys
- Country Surveyed: Luxembourg
- Survey Year: 2023
Output:

{"question": "What is the inflation of Luxembourg in 2023?", "answer": "2.9%"}

*Example of Vague Triplet (Should Return Empty):*
Triplet:
{"entity_1": "luxembourg", "relation": "should maintain", "entity_2": "prudent fiscal policy"}
Text Context:
`[Full example context is omitted...]`
Metadata:
- File Type: OECD Economic Surveys
- Country Surveyed: Luxembourg
- Survey Year: 2023
Output:
{"question": "", "answer": ""}

**Output Format**
Respond in JSON format with "question" and "answer" fields encapsulating the formulated question and its answer.

**Notes**
Ensure questions are specific to the context provided, emphasizing precision and clarity in wording. If no singular answer emerges due to generality, opt for returning an empty dictionary to indicate an unsuitably specific query.

---

### Dataset Generation Prompt: Multi-Hop QA Pairs Generation

You are a benchmark designer creating multi-hop retrieval questions based on three types of multi-hop triplets.

**Input**
- Triplet 1 = ({head1}, {rel1}, {tail1})    <- extracted from Chunk 1
- Triplet 2 = ({head2}, {rel2}, {tail2})    <- extracted from Chunk 2
- Chunk 1: {chunk1}
- Chunk 2: {chunk2}

**Multi-hop Triplets DEFINITIONS**
1. Chain Triplets
- Gurantee: {tail1} == {head2}
- Define A = {head1}, B = {tail1} / {head2}, C = {tail2}
2. Star-shaped Triplets
- Gurantee: {head1} == {head2}
- Define A = {tail1}, B = {head1} / {head2}, C = {tail2}
3. Inverted-star-shaped Triplets
- Gurantee: {tail1} == {tail2}
- Define A = {head1}, B = {tail1} / {tail2}, C = {head2}

**GOAL**
Write ONE natural-language multi-hop question that *requires* evidence from both chunks and answer it succinctly (no full sentences, only essential information).

**ALGORITHM**
1. Decide whether the final answer will be A or C.
- Pick A if you can phrase the question so the solver must:
• hop-1: use (C, rel2) to identify B,
• hop-2: use (B, rel1) to reach A.
- Pick C if you can phrase the question so the solver must:
• hop-1: use (A, rel1) to identify B,
• hop-2: use (B, rel2) to reach C.
2. Write a fluent, specific, and natural question that:
- References the pivot B indirectly (via the opposite hop as above).
- Omits the answer itself.
- Cannot be answered from a single chunk.
- Includes detailed and specific context from the source text chunks. DO NOT just use "according to OECD Economic Survey".
- BAD example: "What is the primary export sector of the country that faces risk from global supply chain disruptions?" (Too vague; could refer to any country)
- GOOD example: "What is the primary export sector of the country that faces risk from global supply chain disruptions in Q3 2021?" (Specific to the context and time frame)
3. Return the answer based on A or C. Ensure the answer precisely matches the facts provided in the context.

**EXAMPLE**
{"entity_1": "forward-looking fuel-tax trajectory", "relation_1": "would reduce", "entity_2": "reliance on combustion-engine cars"}
{"entity_1": "reliance on combustion-engine cars", "relation_2": "drives", "entity_2": "transport-sector emissions"}
*question*: Which forward-looking tax trajectory is proposed to cut the main driver of transport-sector emissions?
*answer*: forward-looking fuel-tax trajectory

**QUALITY CHECKS**
- Pivot-rarity: B must be distinctive ($\geq 2$ meaningful words, not generic terms like "measures", "it", "the company"). If B is too generic, output empty strings for the question and answer.
- Negative-distractor safety: Ask could a system answer your question after retrieving only *one* chunk? If yes, output empty strings for the question and answer.

**OUTPUT**
Respond in JSON format with question and answer only as shown below:
{
"question": "...",
"answer": "..."

```
}
```

---

**Dataset Generation Prompt: Single-Hop QA Pairs Quality Assurance**

**Single-Hop Query Quality Evaluator**

You are an expert evaluator of single-hop queries. Assess each query's quality across two dimensions on a 1-5 scale.

**Assessment Criteria**
1. Clarity (Question and Answer) (1-5)
- 5: Concise, unambiguous wording; answer mirrors clarity
- 4: Minor wording issue but still unambiguous
- 3: Some vagueness but meaning recoverable
- 2: Ambiguities/redundancies hinder understanding
- 1: Unclear or contradictory wording
2. Correctness (vs. Ground-Truth) (1-5)
- 5: Answer matches all facts in chunks; nothing missing
- 4: Correct but one minor fact omitted/loosely paraphrased
- 3: At least half of facts correct; one factual slip
- 2: Major fact missing/misstated/unsupported
- 1: Contradicts or ignores ground truth

**Evaluation Process**
1. Identify reasoning process
2. Assess alignment between query and provided text chunk
3. Evaluate clarity of question and answer
4. Verify factual correctness against ground-truth chunk

**Input**
- query: The single-hop question
- answer: The provided answer
- text chunk: Source text chunk

**Output**
```
{
"score": <average_of_dimension_scores>,
"dimension_scores": {
"clarity": <1-5>,
"correctness": <1-5>
}
}
```

---

**Extracted Triplets Factual Correction Judge Prompt**

You are a strict fact-checking assistant. Verify if the extracted knowledge triplet is factually correct and explicitly supported by the provided text.

**Text:**
- {text}

**Triplet:**
Subject: {triplet['entity_1']}
Relation: {triplet['relation']}
Object: {triplet['entity_2']}

Is this triplet factually correct based ONLY on the text provided?
Answer with **ONLY** 'Correct' or 'Incorrect'. Do not provide any explanation.

---

**Multi-Hop QA Pairs Quality Assurance**

**Multi-Hop Query Quality Evaluator**

You are an expert evaluator of multi-hop queries. Assess each query's quality across three dimensions on a 1-5 scale.

**Assessment Criteria**
1. Reasonableness and Multi-hop Need (1-5)
- 5: Meaningful question requiring all hops; each hop justified
- 4: Reasonable but one hop weakly motivated or could be merged
- 3: Sensible but answerable by single chunk with assumptions
- 2: Forced/trivial question; multi-hop structure unnecessary
- 1: Nonsensical/irrelevant; multi-hop structure meaningless
2. Clarity (Question and Answer) (1-5)
- 5: Concise, unambiguous wording; answer mirrors clarity
- 4: Minor wording issue but still unambiguous
- 3: Some vagueness but meaning recoverable
- 2: Ambiguities/redundancies hinder understanding
- 1: Unclear or contradictory wording
3. Correctness (vs. Ground-Truth) (1-5)
- 5: Answer matches all facts in chunks; nothing missing
- 4: Correct but one minor fact omitted/loosely paraphrased
- 3: At least half of facts correct; one factual slip
- 2: Major fact missing/misstated/unsupported
- 1: Contradicts or ignores ground truth

**Evaluation Process**
1. Identify distinct reasoning hops and assess necessity
2. Check alignment between hops and provided chunks
3. Evaluate clarity of question and answer
4. Verify factual correctness against ground-truth chunks

---

**Input**
- query: The multi-hop question
- answer: The provided answer
- text chunks: Source text chunks

**Output**
{
"score": <average_of_dimension_scores>,
"dimension_scores": {
"reasonableness": <1-5>,
"clarity": <1-5>,
"correctness": <1-5>
}
}

---

**RAG Simulation Prompt: RAG Generator**

You are a {domain} expert. You are given a {domain} question and one or multiple contexts.
Your task is to answer the question strictly based on the these contexts.
You should think step by step and answer the question in a detailed and comprehensive way. Please return the detailed reasoning process in the cot_answer part.

**Requirements:**
- Your answer is short and concise, do not return any other text in the answer part.
- Example 1: "What is the United States' GDP in 2024?"
- Good: "$31.1 trillion"
- Bad: "According to the context, as my knowledge, the answer is $31.1 trillion"
- Example 2: "Who is the president of the United States from 2021 to 2025?"
- Good: "Joe Biden"
- Bad: "The president of the United States from 2021 to 2025 is Joe Biden, according to my knowledge"
- If the question is not related to the context, strictly return "no such info" for answer part. Do not return any other text in such case.

Here are some examples of how to answer based on the given context:
*Example 1:*
Question: What was Apple's revenue in Q2 2023?
Context: [Doc] Apple Inc. reported financial results for its fiscal 2023 second quarter ended April 1, 2023. The Company posted quarterly revenue of $94.8 billion, down 2.5 percent year over year.
cot_answer: The question asks about Apple's revenue in Q2 2023. According to the context, Apple reported quarterly revenue of $94.8 billion for its fiscal 2023 second quarter ended April 1, 2023. This represents a decrease of 2.5 percent year over year.
answer: $94.8 billion

*Example 2:*
Question: What is Luxembourg's approach to public transport?
Context: [Doc] On March 1, 2020, Luxembourg became the first country to make all public transport free, including buses, trains, and trams. This policy aims to reduce traffic congestion and carbon emissions while promoting sustainable mobility solutions across the country.
cot_answer: The question asks about Luxembourg's approach to public transport. According to the context, Luxembourg made all public transport free on March 1, 2020, becoming the first country to do so. This includes buses, trains, and trams. The goal of this policy is to reduce traffic congestion and carbon emissions while promoting sustainable mobility solutions.
answer: Free public transport for all

*Example 3:*
Question: How many homeless individuals received emergency shelter services in Pittsburgh?
Context: [Doc] The City of Pittsburgh allocated CDBG funds to various community programs including affordable housing initiatives. The HOME program supported the construction of 45 new housing units for low-income families.
cot_answer: The question asks about the number of homeless individuals who received emergency shelter services in Pittsburgh. After reviewing the context carefully, I don't see any information about emergency shelter services for homeless individuals or any numbers related to this. The context only mentions CDBG funds for community programs and the HOME program supporting 45 new housing units for low-income families. There is no specific information about homeless emergency shelter services.
answer: no such info

*Example 4:*
Question: What were Smith A O Corp's consolidated sales for the year ended December 31, 2024?
Context: [Doc] In this section, we discuss the results of our operations for 2024 compared with 2023. Our sales in 2024 were $3,818.1 million, a decrease of $34.7 million compared to 2023 sales of $3,852.8 million. Our decrease in net sales was primarily driven by lower water heater volumes in North America, lower sales in China, and unfavorable currency translation of approximately $18 million due to the depreciation of foreign currencies compared to the U.S. dollar, which more than offset our higher boiler sales and pricing actions.
cot_answer: The question asks about Smith A O Corp's consolidated sales for the year ended December 31, 2024. According to the context, the sales in 2024 were $3,818.1 million, which was a decrease of $34.7 million compared to 2023 sales of $3,852.8 million. The context explains that this decrease was primarily due to lower water heater volumes in North America, lower sales in China, and unfavorable currency translation of approximately $18 million.
answer: $3,818.1 million

**Output Format:**
- cot_answer: detailed reasoning process
- answer: concise answer to the question

> **Evaluation Prompt: Judging Prediction and Ground Truth**
>
> You are a fair and strict judger, given prediction and ground truth, your task is to determine if the prediction has the same/highly similar meaning as the ground truth answer.
> Return true if:
> - The prediction and ground truth are semantically identical or highly similar.
> - The prediction provides the same information as the ground truth.
> - If ground truth is included in the prediction consider it a match.
> - Which means if prediction not only contains the ground truth, but also contains other information, it should be considered a match.
> - Example: prediction: "The company's revenue was $50 million in 2023" and ground truth: "$50 million" are considered the same.
> Return false if:
> - The prediction does not match the ground truth in meaning.
> - The prediction is a refusal or does not provide an answer.
> - The ground truth has more specific information than the prediction.
> - If the prediction is a numeric value, it should match the ground truth numerically
> - Example 1: 120,000,000 and 120000000 are considered the same.
> - Example 2: 120,000,000 and 120 billion are considered the same.
>
>
> For your output, you should only answer 'true' or 'false', no extra text.
> **Examples:**
> 1. Prediction: "The company's revenue was $50 million in 2023", Ground Truth: "$50 million", Output: true
> 2. Prediction: "Apple Inc.", Ground Truth: "Apple", Output: true
> 3. Prediction: "I cannot find that information", Ground Truth: "25%", Output: false
> 4. Prediction: "The answer is 42", Ground Truth: "42", Output: true
> 5. Prediction: "The population is around 1 million", Ground Truth: "1,000,000", Output: true
> 6. Prediction: "Tesla", Ground Truth: "General Motors", Output: false

## G EXPERIMENT RESULTS ANALYSIS

### G.1 DETAIL ANALYSIS

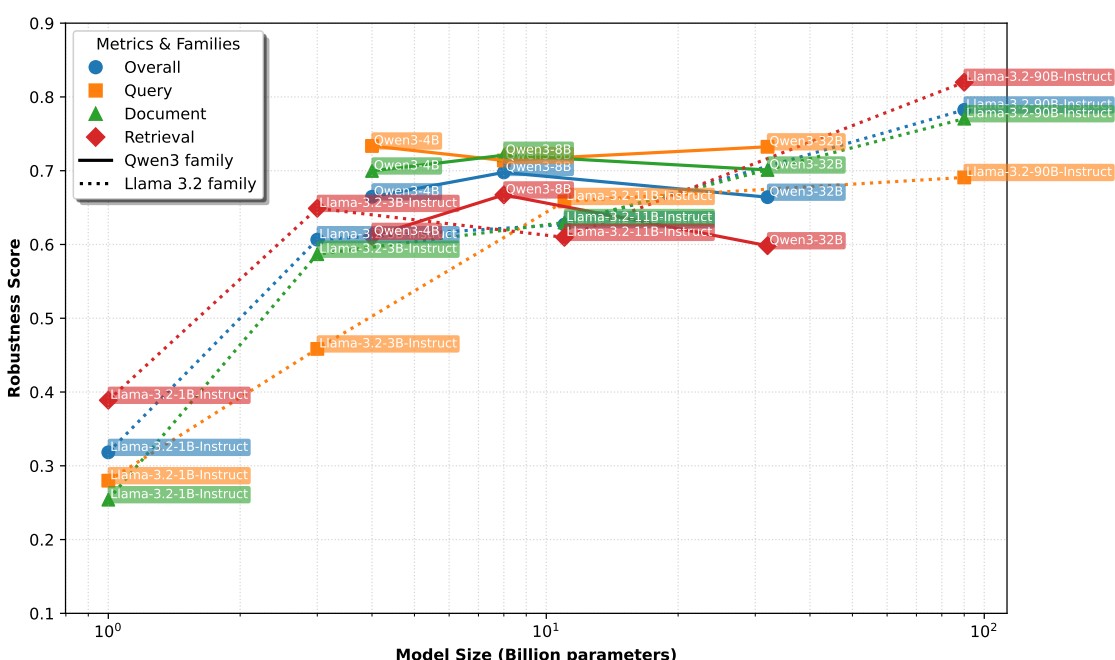

Figure 4: Relationship between the sizes of open-source generators and their robustness scores across various categories. Generally, larger generator sizes correspond to higher robustness scores. However, for Qwen 3 models, robustness scores tend to stay closely across difference parameter sizes

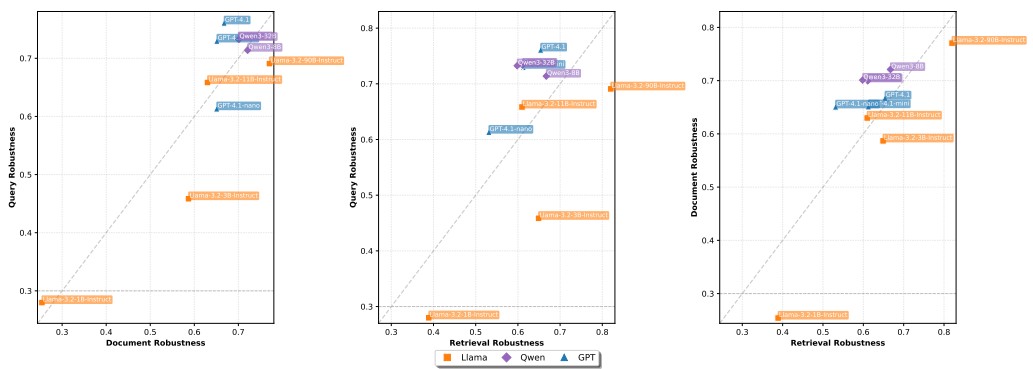

Figure 5: Pairwise relationship between query, document and retrieval robustness. All of these models achieve the balanced robustness across query, document, and retrieval dimensions, while Qwen3 models cluster tightly in the upper-right corner, indicating consistently strong robustness across categories. In contrast, Llama models are more spread out, with smaller ones performing poorly and larger ones improving in document and retrieval robustness but still lagging in query robustness.

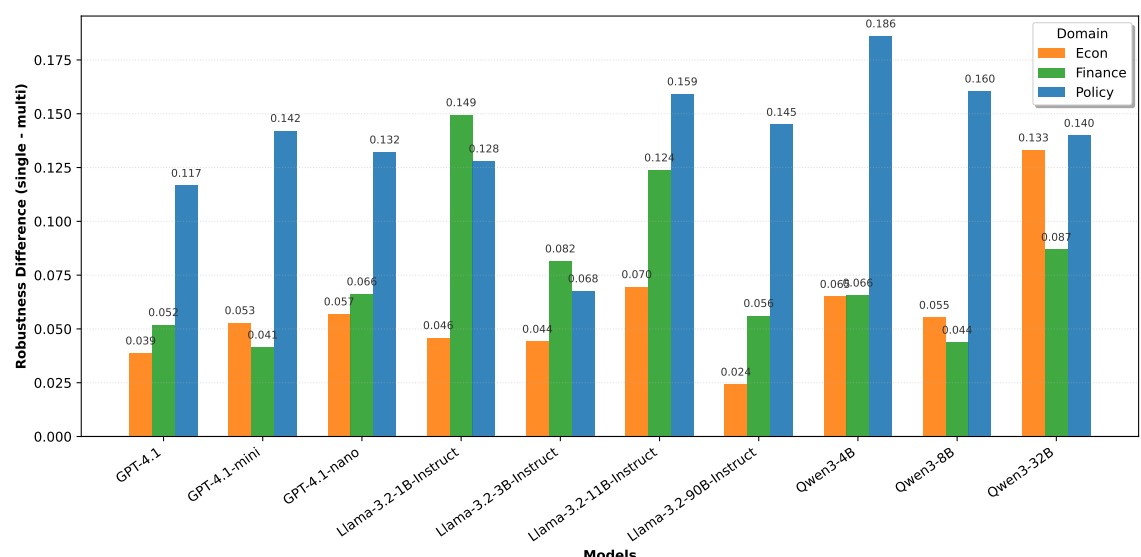

Figure 6: Difference in multi-hop and single-hop robustness scores by domain. Positive robustness scores = single-hop better, negative robustness scores = multi-hop better. Since all of the differences are positive, it clearly shows that RAG systems exhibit lower robustness on multi-hop questions compared to single-hop questions, while the most significant gaps appears in policy domain.

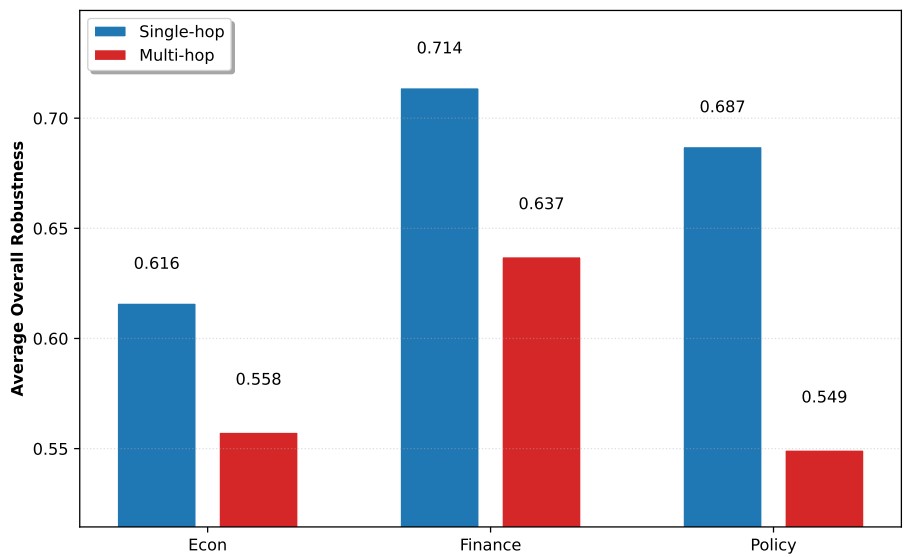

Figure 7: Average overall robustness scores from different domains and question types

G.2 PERTURBATION-SPECIFIC ROBUSTNESS

Figure 8: Average robustness score in different query perturbations vs. all types of documents.

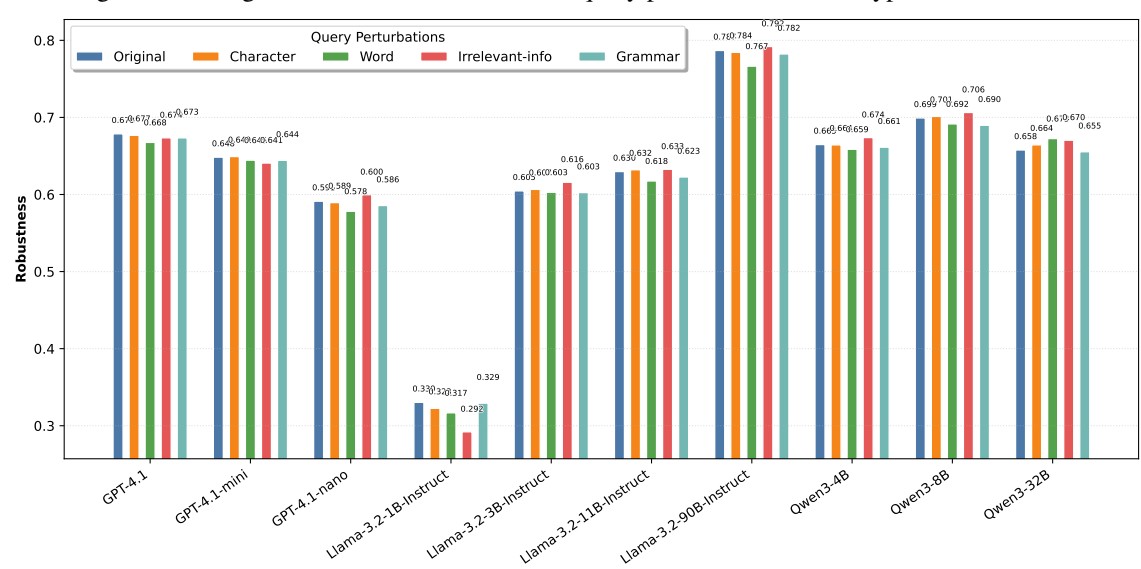

Figure 9: Average robustness score in different document perturbations vs. all types of queries.

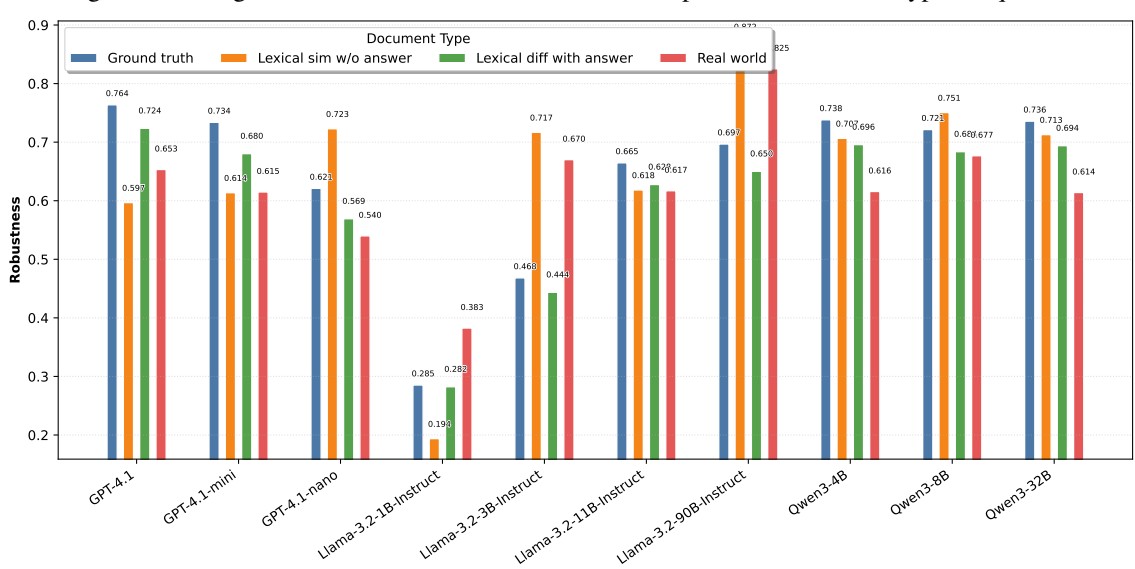

### G.3 SIGNIFICANCE TESTS

| Model 1 | Model 2 | Z-score | P-value |
|---------|---------|---------|---------|
| **Qwen3 vs GPT** | | | |
| GPT-4.1 | Qwen3-32B | 6.312 | 0 |
| GPT-4.1 | Qwen3-4B | 5.987 | 0 |
| GPT-4.1 | Qwen3-8B | -15.300 | 0 |
| Qwen3-32B | GPT-4.1-mini | 11.772 | 0 |
| Qwen3-32B | GPT-4.1-nano | 46.655 | 0 |
| Qwen3-4B | GPT-4.1-mini | 12.096 | 0 |
| Qwen3-4B | GPT-4.1-nano | 46.978 | 0 |
| Qwen3-8B | GPT-4.1-mini | 33.362 | 0 |
| Qwen3-8B | GPT-4.1-nano | 68.138 | 0 |
| **Qwen3 vs Llama** | | | |
| Llama-3.2-90B | Qwen3-32B | 79.336 | 0 |
| Llama-3.2-90B | Qwen3-4B | 79.016 | 0 |
| Llama-3.2-90B | Qwen3-8B | 57.961 | 0 |
| Qwen3-32B | Llama-3.2-11B | 23.295 | 0 |
| Qwen3-4B | Llama-3.2-11B | 23.620 | 0 |
| Qwen3-8B | Llama-3.2-11B | 44.860 | 0 |
| **GPT vs Llama** | | | |
| GPT-4.1 | Llama-3.2-11B | 29.599 | 0 |
| Llama-3.2-11B | GPT-4.1-mini | -11.529 | 0 |
| Llama-3.2-11B | GPT-4.1-nano | 23.402 | 0 |
| Llama-3.2-90B | GPT-4.1 | 73.107 | 0 |
| Llama-3.2-90B | GPT-4.1-mini | 90.924 | 0 |
| Llama-3.2-90B | GPT-4.1-nano | 125.065 | 0 |

Table 8: Pairwise two-proportion z-tests comparing models' overall robustness scores. Every pair of p-value is less than 0.05, indicating our results' high statistical significances.

## H THE USE OF LLMS

We acknowledge the use of LLMs in the writing of this paper. They were used to check grammar and improve sentence clarity. In addition, LLMs were utilized in our data generation pipeline and during the evaluation stage. These uses are explicitly described in the corresponding sections.

