# OpenReview forum: "RARE: Retrieval-Aware Robustness Evaluation for Retrieval-Augmented Generation Systems"
_ICLR.cc/2026/Conference — Submitted to ICLR 2026_

### Official Review · Reviewer_a38J · 2025-10-31

**Soundness:** 3
**Presentation:** 3
**Contribution:** 2
**Rating:** 4
**Confidence:** 3

**Summary:**

This paper introduces Retrieval-Aware Robustness Evaluation (RARE), a comprehensive framework designed for evaluating retrieval-augmented generation systems. Without requiring manual curation, RARE automatically constructs time-sensitive benchmarks via knowledge graph extraction and traversal, and systematically assesses robustness under query, document, and real-world retrieval perturbations. Extensive experiments across financial, economic, and policy domains with over 48,000 queries demonstrate that RARE effectively reveals critical limitations in RAG systems, particularly in multi-hop and domain-specific scenarios.

**Strengths:**

1. The authors introduce Retrieval-Aware Robustness Evaluation (RARE), a comprehensive framework designed for evaluating retrieval-augmented generation systems.

2. Extensive experiments across financial, economic, and policy domains with over 48,000 queries demonstrate that RARE effectively reveals critical limitations in RAG systems, particularly in multi-hop and domain-specific scenarios.

3. The analysis dimensions are thorough and insightful.  The paper not only presents overall performance comparisons but also compares robustness under query perturbations, document perturbations, and real retrieval scenarios.

**Weaknesses:**

1.  Insufficient technical details in the knowledge graph construction process: The paper does not provide enough description of the core steps of knowledge graph construction, particularly the specific implementation of relation normalization (detailed process using E5-Mistral-7B-Instruct), the quality control mechanisms for handling extracted conflicting or incorrect triples, and how to manage entity alignment and conflict resolution during cross-document knowledge graph merging.

2.  The comparison across unique dimensions is not clearly explained.  In terms of time-sensitivity (time-sens), multi-hop questions (MH), and cross-document retrieval (Dynamic), the improvements are not significantly different from existing datasets, and the innovation in these dimensions is insufficient.

3.  Limited coverage of perturbation types: The current perturbation types mainly focus on surface-level and semantic-level changes but lack factual perturbations (e.g., inserting contradictory information into documents), structural perturbations (e.g., changing document order or format), and handling of multimodal documents (e.g., cases where tables and text are mixed in images).

4.  Practical guidance could be strengthened: The experiments identified several important phenomena (e.g., the vulnerability of multi-hop queries), but there is a lack of in-depth analysis of the underlying causes of these phenomena and no specific system improvement suggestions based on the findings.

**Questions:**

1. The core innovation of the paper relies on large models like GPT-4 for knowledge graph triple extraction and QA pair generation.      Considering the large scale of the benchmark datasets, what are the specific computational costs and time overheads of this automated process?      Have more cost-effective alternatives been explored?      Providing estimates and discussion on this aspect would help the community replicate and extend this work.

2. On the realism and intensity balance of perturbations: In document perturbations, "lexically similar but answerless" is achieved by directly deleting the answer sentence, which may be too extreme and easy to identify in real retrieval scenarios.      Have more natural and subtle perturbation methods been considered?      Additionally, how can it be ensured that different types of perturbations have roughly comparable "interference intensity" to avoid overly harsh or lenient evaluation of certain perturbation types?

4. On the fairness of model comparison and prompt engineering: Experiments compared various open-source and closed-source models, but different models may have different sensitivities and optimization needs for prompts.      Were the unified prompts used in the paper sufficiently optimized or validated for each evaluated model?      If not specifically optimized, could this potentially lead to an underestimation of some models’ performance?

5. On the phenomenon that "model robustness does not strictly increase with scale": The paper observes an interesting phenomenon where Qwen3-32B shows lower robustness than the smaller Qwen3-8B and 4B models, attributing it to "architecture design and training methods."      This explanation could be more specific and detailed.      Can more nuanced hypotheses be proposed?

6. On the basis for quality assurance threshold selection and data bias analysis: In the quality assurance stage, Claude 3.5 was used to score generated QA pairs, with a retention threshold set at "all dimension scores above 6."      What considerations were behind this specific threshold?      Can the score distribution of filtered-out QA pairs be provided, analyzing which dimensions mainly had deficiencies?      This would help readers understand the quality boundaries of the dataset and potential biases.

If the authors can address all of the concerns, I am willing to consider raising my rating.

---

> ### Author Response · Authors · 2025-11-21
>
> # Responses
> ## Response for weakness #1: details about knowledge graph construction pipeline
>
> Thank you for your question regarding the details of the knowledge graph construction process.
>
> Our pipeline performs relation normalization by using e5-mistral-7b-instruct to generate embeddings for extracted relations. New relation labels are mapped to existing canonical ones when their cosine similarity exceeds a chosen threshold. To determine this threshold, we sampled relation pairs, evaluated multiple candidates, and ultimately selected **0.9** based on human majority voting. Domain-specific prompts are used throughout this process to ensure schema consistency.
>
> For quality control, we require GPT-4.1 to output all *source sentences* from the original document chunk used to generate each triplet. During validation, we check that these source sentences appear verbatim in the chunk (normalized exact match), discarding any triplets whose sources are hallucinated or unverifiable.
>
> Furthermore, entity alignment across documents is achieved through a text-normalization procedure that includes lowercasing, punctuation removal, and domain-specific stripping of corporate suffixes (like “Inc.”, “Corp.” for finance). Cross-document merging is then performed by combining individual graphs using NetworkX (such as grouping by GICS sectors in finance) to unify nodes sharing the same normalized entity identifiers.
>
> All of these details above have been added to the Appendix in the revised version (Section 3.2)
>
> ## Response for weekness #2: dataset dimensions and novalty
> We appreciate the reviewer’s question.  First, we would like to emphasize that our dataset is substantially different from prior benchmarks (shown in Table 1). While most existing time-sensitive or multi-hop benchmarks rely on Wikipedia or news and are either static or only partially auto-updated, RARE-Set is built from up-to-date SEC 10-K filings, OECD economic surveys, and HUD CAPER policy reports, yielding expert-level, domain-specific and time-sensitive queries that cannot be reliably answered from a model’s memorized Wikipedia or general web knowledge.
>
> Furthermore, our main contribution is not just ticking the boxes of time-sensitivity, multi-hop, and dynamic/cross-document retrieval, but providing a unified, fully automatic synthesis pipeline (RARE-Get) that can continuously build new benchmarks over evolving, domain-specific corpora. From the corpus-selection perspective, our pipeline is designed to work on any text- or table-based corpus. From the generation perspective, our KG-driven pipeline enforces multi-hop *cross-document* reasoning patterns that cannot be answered from a single chunk or a model’s memorized knowledge, while still scaling to tens of thousands of high-quality QA pairs without human interference. We believe this end-to-end, dynamic, KG-based synthesis pipeline is where our most significant innovation lies.
>
> ## Response for weakness #3: coverage of perturbation types:
> Thank you for your suggestions, and we agree that there are more diverse perturbation methods that could further enhance the robustness evaluation of RAG systems, such as different document structures. Our goal in this work is to propose a *dynamic testbed* for robustness evaluation in LLM-based RAG systems, including both automatic benchmark creation and robust metrics along three dimensions. In this paper, we therefore focus on systematically analyzing RAG systems' robustness on three different dimensions, including queries, documents, and real-world retrieval. Instead of covering all potential perturbations in this work, we would like to leave others, such as structural document perturbations, to future work.
>
> For multimodal perturbations, since our focus is to automatically create large-scale **text** datasets for RAG evaluation and to examine LLM-based RAG systems, rather than the robustness of multimodal models. In addition, synthesizing desired perturbations for multimodal content is far less controllable and much more difficult to ensure quality, compared with text-only content, which is a new area to be explored.
>
> For factual perturbations, because our selected documents come from dynamic and often niche domains, LLMs tend to perform poorly in these areas. According to our experiment, in a random sample of 6000 questions, GPT-4.1 could answer only **10.7%** without retrieval, indicating current models’ limitations in such domains. Therefore, any perturbations requiring LLM generation cannot guarantee generation and verification quality, especially when we want to scale dataset up.

---

> > ### Author Response · Authors · 2025-11-21
> >
> > ## Response for weakness #4 and question #5: further explanation about causes (training and architecture perspective) of low robustness score
> >
> > Regarding the in-depth analysis you mentioned in both the questions and the weaknesses, specifically, explaining from the training or model-architecture perspective why these models lack robustness. We acknowledge that our statement, “All of these findings highlight the decisive roles of architectural design and training methodology,” is merely a conjecture. We have revised this sentence to make our analytical conclusions more rigorous.
> >
> > ## Response for question #1: Computation/time costs for dataset generation and evaluation
> >
> > Our framework is explicitly designed to minimize both cost and uncertainty from LLM calls by relying as much as possible on traditional, easily verifiable components.
> >
> > We run all open-source models using 16 NVIDIA L40S GPUs, and it takes about five days end-to-end (mentioned from Line 375 to 377). The only substantial monetary cost comes from three closed-source GPT-series models used in a few key stages (GPT 4.1 for KG triple extraction + QA generation, and evaluation for three GPT models), totaling roughly **$3400** in API fees. This cost is significantly reduced compared to an end-to-end LLM-based dataset generation pipeline because over **90%** of our API tokens are input tokens, which are much cheaper than output tokens. Also, KG construction is a one-time expense: once built, the graph can be reused to regenerate or extend the benchmark without incurring comparable additional LLM costs, making the framework practical to replicate and extend.
> >
> > ## Response for question #2: straightforward perturbation and "interference intensity" of perturbations difficulties
> > Deleting the answer sentence may look extreme in isolation, but it is meant to simulate common real-world issues such as missing pages, aggressive truncation, or incomplete coverage in the retrieved context. We still have other more natural surface-level and semantic perturbations (e.g., paraphrasing, back-translation) that preserve the answer but make matching harder.
> >
> > For your concern about keeping similar “interference intensity” across different perturbations, our benchmark’s goal is not to ensure that all perturbations have identical intensity, but to make overall scores (and sub-scores) more distinguishable. As you can see in experiment analysis in Appendix, our results do show a significant difference and faithfully reflect different models’ performances.  Different robustness evaluations should have different levels of difficulty, since this better reflects systems’ actual capabilities under varying perturbations. Similar to human exams, if all question types had the same difficulty, it would be impossible for a teacher to identify which areas a student does not fully understand. Our evaluation follows this same principle.
> >
> > ## Response for question #3: optimal prompt for different models
> > For open-source models like Qwen, we use their recommended settings and input formats. For closed-source models, finding the most “optimal” prompt is hard, and our RAG generator’s prompt follows principles from the official GPT-4.1 Prompt Guide to ensure model performance. Furthermore, to better reflect real-world use, we don’t always fully optimize prompts since most users cannot provide the optimized prompts for different models. Using a unified prompt keeps comparisons fair and also tests models’ adaptability, even if that isn’t our main goal.

---

> > ### Author Response · Authors · 2025-11-21
> >
> > ## Response for question #5: threshold selection and statistics for qualit-assurance by dimensions
> > First, we need to clarify that we actually use 1-5 scale from Claude 3.5 and retain QA pairs only if **all dimensions** score strictly above 3 (not 6). This threshold is chosen by human calibration. In a 100-sample check, only around 40% of items with all dimensions threshold = 2 were judged acceptable by annotators, while this jumped to 80% for threshold = 3, with only limited gains (85 to 90%) when further tightening the threshold to 4 or 5. So threshold = 3 offered the best trade-off between final quality and dataset size.
> >
> > For your question regarding how many generated questions are filtered out. The acceptance rate in our LLM-based quality-assurance stage is about 80.31% (so **19.69%** of the dataset is filtered out). In addition, here're the score distribution of different categories for filtered QA pairs:
> >
> > | Dimension | Dataset Type | Score = 3 | Score = 4 | Score = 5 |
> > | :---- | :---- | :---- | :---- | :---- |
> > | Reasonableness | Multi-Hop | 20.8% | 63.1% | 16.1% |
> > | Correctness | Multi-Hop | 17.3% | 68.5% | 14.2% |
> > | Clarity | Multi-Hop | 13.6% | 47.9% | 38.5% |
> > | Correctness | Single-Hop | 11.8% | 62.9% | 25.3% |
> > | Clarity | Single-Hop | 3.6% | 12.1% | 84.4% |
> >
> > Across both multi-hop and single-hop datasets, most QA pairs score = 4 in their respective dimensions, with single-hop clarity being a clear standout at 84.4% achieving a score of 5. It also proves our pipeline could consistently generate high-quality evaluation benchmarks. However, certain multi-hop dimensions, such as correctness and reasonableness, still show room for improvement (relative lower score = 5 distribution rate compared with multi-hop clarity).

---

### Official Review · Reviewer_gD5y · 2025-11-01

**Soundness:** 2
**Presentation:** 3
**Contribution:** 3
**Rating:** 4
**Confidence:** 4

**Summary:**

This paper introduces a framework for evaluating RAG system robustness under realistic perturbations. RARE addresses critical gaps in existing benchmarks by simultaneously examining three key dimensions: dynamics (handling changing information), query complexity (single-hop to multi-hop reasoning), and robustness measurement (quantifying performance degradation). The framework consists of RARE-Get (automated KG-driven synthesis pipeline), RARE-Set (questions spanning multiple documents and complexity levels), and RARE-Met (retrieval-conditioned robustness metrics). Evaluation reveals fragility in current RAG systems.

**Strengths:**

- The proposed method is highly scalable: RARE-Get uses KG-driven synthesis to automatically generate multi-hop questions without manual curation, focusing on specialized, time-sensitive corpora appropriate for real-world RAG applications.
- This paper proposes well-defined metrics: RARE-Met distinguishes memorization from retrieval-based reasoning and tests robustness across query perturbations (typos, paraphrasing) and document perturbations (lexical/answer variations). Moreover, its evaluation presents useful insights: model size doesn't guarantee robustness, multi-hop queries underperform single-hop, and domain factors significantly impact performance.

**Weaknesses:**

- The paper does not analyze error propagation which harms its reliability. RARE-Get chains multiple LLMs (GPT-4.1 for extraction and generation, Claude variants for filtering and evaluation), where errors compound at each stage. The paper omits failure rates, data discarded during quality checks, and how extraction errors corrupt downstream question generation.
- The paper assumes generated questions require multi-hop reasoning without validation—questions may be answerable through single-chunk retrieval despite their structure. The robustness definition is debatable: it penalizes models that follow perturbed context, but robust systems could instead detect and reject unreliable retrieved information, yielding opposite interpretations of the same behavior.

**Questions:**

Could you comment on the failure rate or amount of data discarded during the LLM-based quality assurance step? What percentage of extracted triplets are factually correct? For instance, sample 100-200 triplets and verify against source documents can give a better sense of the raw quality of initial LLM-based generation and how errors propagate through the pipeline.

---

> ### Author Response · Authors · 2025-11-21
>
> # Responses
> ## Response for weakess #1 and question #1: error propagation and extracted triplets factual accuracy
> We acknowledge that multi-stage LLM pipelines risk error propagation, as upstream hallucinations can influence downstream steps. Therefore, we design a final-stage end-to-end data quality check using LLM-as-a-judge. Instead of filtering errors after each stage of the pipeline, our approach minimizes LLM usage to ensure efficiency while also maintaining the final data quality.
>
> Specifically, our LLM-based quality-assurance stage had an acceptance rate of **80.31%**, with only **19.69%** of triplets rejected. To assess factual correctness, we randomly sampled 1000 triplets and evaluated them with Claude Sonnet 4.5 (since GPT-4.1 generated the triplets). The results show that **87.1%** of the triplets were factually correct relative to their source chunks, indicating that the pipeline reliably captures information from the original documents. All of the error analysis has been added to the revised version (starting from Line 298)
>
> Here’s the prompt we use for judging whether a triplet is factually correct (also available in Appendix):
> ```
> You are a strict fact-checking assistant. Verify if the extracted knowledge triplet is factually correct and explicitly supported by the provided text.
>
> Text:
>    {text}
>
> Triplet:
>    Subject: {triplet['entity_1']}
>    Relation: {triplet['relation']}
>    Object: {triplet['entity_2']}
>
> Is this triplet factually correct based ONLY on the text provided?
> Answer with ONLY 'Correct' or 'Incorrect'. Do not provide any explanation.
> ```
> ## Response for weakness #2: multi-hop questions construction and robustness definition
> ### Sub-question #1: whether multi‑hop questions may actually be single‑chunk answerable.
> In our benchmark, we explicitly try to eliminate such cases through multiple steps below:
>
> 1. **KG-based multi-hop construction**: multi-hop questions are created from chain, star, or inverted-star patterns of KG triplets, with each triplet linked to a different supporting chunk. After traversing the KG, we keep only those groups whose answer\_chunk\_ids are all distinct, ensuring that the evidence for each multi-hop question spans multiple chunks.
> 2. **Negative-distractor safety in generation:** when generating the final question based on the triplets and chunk, our prompts include a "negative‑distractor safety" rule (mentioned in Line 256-257, actual prompt is available in Line 1071-1072): if the LLM can answer with a single chunk, it must return an empty question/answer, and we discard that item.
> 3. **Independent multi-hop quality assurance:** during the quality assurance part, we run an independent multi‑hop quality check: we only keep QA pairs where
>    1. `answer_chunk_ids` has length \> 1
>    2. An evaluator model scores the "Reasonableness & Multi‑hop Need" dimension (definition is available on Line 1151 and beyond) larger than or equal to 3, explicitly removing questions judged to be trivially single‑hop despite their surface structure.
>
> ### Sub-question #2: whether our robustness metric unfairly penalizes models that reject unreliable context
>
> Thank you for raising this point. Our robustness definition ***does*** account for systems that correctly detect **and** reject unreliable retrieved information. In our framework, robustness is *not* limited to following perturbed context; instead, the scoring criteria explicitly distinguish between answer-bearing and non-answer-bearing retrieval.
>
> When the retrieved documents ***don’t contain*** the answer (such as lexical-similar but no answer or some real-world retrieval chunks), we treat **either** producing the correct answer **or** issuing a safe refusal as robust behavior (last two rows of Table 3). This directly captures the scenario you mentioned: a model that identifies unreliable/misleading retrieval and declines to rely on it is rewarded, not penalized.
>
> We only define the output as robust iff the retrieval is intended to be correct and answer-bearing (like ground truth docs, lexical difference but with answer). In these settings, rejecting the context would indicate a failure to use reliable information.
>
> As the summary, the definition already incorporates both dimensions of robustness:
> 1. Using correct retrieval when retrieval is trustworthy.
> 2. Detecting and rejecting when retrieval is unreliable.

---

### Official Review · Reviewer_nXfs · 2025-11-01

**Soundness:** 3
**Presentation:** 3
**Contribution:** 3
**Rating:** 6
**Confidence:** 3

**Summary:**

This paper focuses on evaluating existing Retrieval-Augmented Generation (RAG) systems. It first identifies the limitations of current evaluation frameworks and then introduces a novel approach for automatically generating complex evaluation data. Specifically, the paper presents RARE-Get, a synthesis pipeline that constructs time-sensitive evaluation data via knowledge graphs; RARE-Set, a multi-domain dataset created using this pipeline; and RARE-Met, an evaluation metric designed to assess RAG performance under query and document perturbations. Experimental results demonstrate that current RAG systems exhibit fragility under certain perturbations and that robustness does not always increase proportionally with model size.

**Strengths:**

- This paper identifies key limitations in existing RAG evaluation datasets and introduces a dynamic data generation pipeline that leverages knowledge graphs to automatically construct both single-hop and multi-hop questions. The proposed pipeline has strong potential value for the community, as it can generate diverse and complex datasets from unstructured data. In addition, the proposed dataset **RARE-Set** and the robustness evaluation metric contribute useful resources for assessing RAG systems.
- The paper is well-written, and the analysis is thorough, supported by extensive experiments.

**Weaknesses:**

Although the paper includes experiments with various LLMs, the baseline methods are limited to standard RAG setups. Recent state-of-the-art RAG variants, such as adaptive or noise-robust models, are not included, which makes it difficult to fully assess the effectiveness of the proposed benchmark and evaluation metric. Incorporating results from these stronger baselines would significantly enhance the rigor and credibility of the evaluation.

**Questions:**

N/A

---

> ### Author Response · Authors · 2025-11-21
>
> # Response for the weakness
> We appreciate the reviewer’s acknowledgement of our contributions on introducing a dynamic data generation pipeline that can generate diverse and complex datasets. We also thank the reviewer for suggesting the inclusion of adaptive and noise-robust RAG variants. Our primary goal, however, is to introduce **RARE including a dynamic dataset creation pipeline, domain-specific benchmarks, and evaluation metric** for retrieval-aware robustness, rather than to optimize a specific RAG algorithm. For this reason, we intentionally used strong but *standardized* RAG pipelines, covering top MTEB retrievers and diverse LLM generators, to ensure fairness and reproducibility across the three expert-level domains.
>
> We agree that including an advanced noise-robust RAG system would further contextualize our results. These methods, however, often rely on domain-specific training or additional supervision that may confound our domain-specific robustness evaluation settings. Nonetheless, in the camera-ready version we will incorporate a representative robust RAG baseline and expect the central findings, particularly the consistent fragility under multi-hop and perturbation stress, will remain unchanged.

---

> > ### Comment · Reviewer_nXfs · 2025-11-26
> > **Response to Authors**
> >
> > I appreciate the authors’ response and their commitment to revising the manuscript. I do not have other questions. I will retain my current scores.

---

### Meta-Review · Area_Chair_QLU3 · 2026-01-03

**Summary:**

Retrieval-Augmented Generation (RAG) is widely used as a technique to augment LLM’s generation. This paper aims to evaluate the robustness of RAG under adversarial perturbations to queries and documents. The paper finds that RAG systems are very sensitive to perturbations. Additionally, they have lower robustness on multi-hop queries compared to single-hop queries.

**Reviewer Concerns:**

The reviewers generally agree that the problem studied in this paper is important. The paper also identifies key limitations in existing RAG evaluation datasets and introduces a dynamic data generation pipeline to address the limitations. The paper is also easy to follow. The reviewers also note the limitations of the paper, which can be addressed before publication, such as (1) evaluating state-of-the-art RAG setups rather than being limited to standard RAG, (2) limited coverage of perturbations, e.g., the paper can evaluate diverse attacks. In general, as an evaluation paper, the experiment of this paper is very limited. The paper may perform a more comprehensive evaluation to bring more insights. The paper may also pay attention to the page limit, e.g., the submission is limited to 9 pages rather than 10.

**Reviewer Scores:**

The reviewers may not change the scores.

---

### Decision · Program_Chairs · 2026-01-26

Reject